# The Preacher as Artist: An Exploration of Sermon Creation as Art-Making

Ruthanna B. Hooke

Virginia Theological Seminary, Alexandria, VA 22304, USA; rhooke@vts.edu

**Abstract:** Preaching is one of the most creative things a pastor does. This essay explores how a theology of creativity, the imagination, and the arts can encourage preachers to embrace proclamation as creative work. The invitation to preachers to engage their creativity and imagination in preaching rests on the theological claim that creativity is intrinsic to human beings as made in the image of God the Creator. To create is to realize a core human vocation and to deepen knowledge of God. The imagination is a primary avenue to such knowledge, since the imagination is a faculty that allows for a holistic grasp of realities both seen and unseen. An artistic approach to preaching is appropriate in that art functions in similar ways to preaching: like preaching, art explores the depths of human existence, creates wholes out of fragments, and makes connections between seemingly disparate phenomena. The dispositions of the artist are vital for preachers, especially the courage and risk-taking required in art-making as a venture into the unknown. These functions of art and qualities of the artist lead to reflections concerning the particular challenges involved in being a Christian artist, and to the role of beauty in the knowledge of God and hence in preaching.

**Keywords:** art-making; aesthetics; beauty; creativity; risk; the self; vocation

## 1. Introduction

One of the most popular and fruitful assignments in my Introduction to Preaching class requires students to create an artistic interpretation of a Scripture passage on which they are preaching. In fulfilling this assignment, students have written poems, painted pictures, made collages, written pieces of music, created dances, and carved sculptures. Both the students and I are surprised by what they create, which is often more profound and insightful than their sermons are. I observe that the students feel freed up by the assignment, able to tap parts of their psyches and wisdom that a more standard process of sermon creation does not always make available. Moreover, the freedom and creativity of these artworks often flow back into the sermons on which they are working, enlivening and deepening the sermons themselves.[1]

This essay aims to think through what is happening in these creative enterprises, why students experience them as delightful and freeing, and how and why artistic activities open new depths in the students' sermons. I will argue that giving permission for preachers to think of their sermons as artistic enterprises, and to think of themselves as artists, gives them access to a powerful means of bringing vitality to sermons, allowing them to discover or rediscover delight and freedom in sermon creation. I will explore how what preachers do is similar to what artists do, and how this comparison offers guidance for *how* preachers can engage the preaching task.

Making this argument is somewhat controversial, in that there are theological objections to describing preaching as an artistic activity. Some of these objections have to do with broader questions about how and in what way Christians can be artists, and some are more narrowly concerned with labeling preaching as artful. The concern about whether and how Christians can be artists arises in part from a perceived conflict between the Christian's obedience to the authority of Scripture, doctrine, and tradition, and modern notions of the artist as an autonomous genius, not bound to orthodoxy, tradition, or community (Begbie 1991, pp. 197–98;

Hart 2014, p. 22). There is also a concern that aspects of the arts are antithetical to Christian ethics; as Frank Burch Brown notes, "Christian theologians have insisted time and again that the association of the arts with the senses, with self-indulgence, with worldly entertainment, and with things purely fictitious can render the arts either trivial or dangerous" (Brown 2000, p. 32). As Brown suggests, the artist's reliance on the imagination puts them under suspicion of trafficking in untruth, since the imagination itself is sometimes seen as morally ambiguous. Trevor Hart points out that, in Scripture, the human capacity to imagine, and the works of human hands that result from these imaginings, can lead readily to idolatry (Hart 2014, pp. 47–50).[2] Michael Bauer notes the ways that these misgivings about art-making in relation to Christian faith manifest themselves in restrictions placed on artistic expression in churches, from a prohibition of dance to the forbidding of certain musical instruments or visual images (Bauer 2013, p. 37).[3]

Even if a convincing case is made for Christian artistry, objections arise to viewing preachers as artists, or the crafting of sermons as akin to art-making. For theologies of preaching that conceive of preaching as a form of the Word of God, an event of divine-human encounter, it would seem crucial that preachers are bound by the authority of Scripture and tradition; to view their work as artistry could open them to a perilous heterodoxy. If the sermon is an instance of God's self-revelation, it could be argued that describing the preacher as an artist puts too much emphasis on the preacher's skills, rather than on God's grace, as the agent that makes preaching a revelatory event.[4] Finally, especially in Protestant thought, there is the conviction that the Word, rather than the arts, is the appropriate means of divine self-revelation, thus setting preaching over against the arts. This understanding tends to rely on the notion that the preaching of the Word is directed toward securing intellectual assent, and that such assent is the basis of faith, rather than esthetic experience or appeals to the imagination (Graham 2012, p. 92).

In response to broader concerns about Christian artistry, various theological accounts of human creativity have been developed, showing how such artistry can be consonant with and expressive of Christian faith. These descriptions of the relationship between art-making and Christian faith provide a foundation for discussing preaching as a form of Christian art-making, demonstrating that what preachers are doing is similar to these descriptions of Christian artistry. These accounts suggest how Christian preaching can properly be understood as artistic; they also suggest ways of practicing this artistry that participate faithfully in the revelatory event of preaching.

## 2. Artistry and Christian Discipleship in Art-Making and Preaching

Theological accounts of the relationship between art-making and Christian discipleship make the claim that artistry is fundamental to our relationship with God. Art-making is central to the divine-human relationship in part because the human urge to create is innate, as it is rooted in our being as created by God. Genesis 1:26–27 describes humans, alone among created beings, as made in the image of God. Many theories have been advanced to explain what this claim means. Being made in the *imago Dei* has been taken to refer to the human capacity to reason, or to speak, or to make signs, or to have free will. However, the Genesis passage itself, in which this description is bestowed on humans, suggests that one of the primary characteristics of being made in God's image is that humans are creative beings. The claim that humans are created in the image of God is made in the midst of the account of God's creation of the universe. Perhaps the most salient characteristic of God, as God is introduced in this passage at the very beginning of Scripture, is that God is creative. As the narrative abundantly illustrates, God is the Creator of all that is; an overflowing creativity is of God's essence. In this context, to describe humans as made in God's image is to suggest that humans are imbued likewise with a share of this creative energy. Indeed, Makoto Fujimura notes that God invites humans to be creative from the beginning of Genesis, when God asks Adam to name all of the other creatures (Fujimura 2020, pp. 96–97).

God's creativity and human creativity are not fully analogous, of course. To begin with, God creates ex nihilo, whereas humans create from materials given in the world. Moreover, God does not need to create in order to realize Godself, but there is a sense in which humans need to create in order to discover who they are, to find their place in the world. God's creativity is a full expression of God's identity, but human creativity is always only a partial expression of our identity, and indeed reveals our unfinishedness. Despite these differences, however, the desire to create is intrinsic to both God and humans. Part of what makes the art-making assignment in homiletics class delightful to my students is that it allows them to fulfill this inborn urge to create.

It is not only that art-making fulfills our inherent desire to create, but also that making art is one of the primary ways that we respond to and participate in God's own working in the world. Jeremy Begbie defends art-making as an appropriate expression of Christian identity by arguing that human artistry is vital to the fulfillment of God's redemptive purposes. Begbie claims that God calls humans to be "secretaries" and "priests" of creation; while all of creation gives praise to God, it is through the human creature that this praise finds a voice and becomes articulate (Begbie 1991, p. 177). As Douglas John Hall describes this human vocation, "here the creation gathers itself and addresses the One whose glorious Word brought it into being, word answering Word" (Hall 1986, p. 204). In engaging in creative activity, humans participate in the work of Christ the Word made flesh, who in his priestly role offers the world to God. Human creativity likewise offers the world to God; hence, "human creativity is supremely about sharing through the Spirit in the creative purpose of the Father as he draws all things to himself through his Son" (Begbie 1991, p. 179). Creativity, on this account, is not only a response of praise to the glories of creation but participates in God's redemptive purposes for creation. The human vocation to create is an invitation to "take the materials given to hand and mind and develop (elaborate), reconfigure, and, indeed, heal them in ways that praise the Creator". This vocation imitates that of Christ, who likewise entered into the things of creation in order to heal and make them new; thus, humans are invited to join with Christ in "strenuous engagement with creation, to extend and elaborate creation's praise, in anticipation of the restoration of all things that has already been embodied in him" (Begbie 2018, p. 142).

Human beings experience this God-given vocation to create when they perceive the beauty of creation, which prompts them to want to create in response to the beauty perceived. As Elaine Scarry notes, to behold beauty is to want to reproduce it, to create (Scarry 1999, p. 4). There is something in us that wants to capture, articulate, recreate the beauty that we encounter, and not just to passively behold it. Wonderfully, through the process of creating, we discover more fully the beauty we have not made. We create in response to the beauty given us, and then, through this "strenuous making, we discover more fully what we have not made" (Begbie 2018, p. 23). Thus, between "given" beauty and "generated" beauty there is no conflict. It is not the case that artists are imposing order on an otherwise chaotic and meaningless world, but rather, through their artistry, they are discovering more fully the beauty that God created and continues to create, and that is already embedded in all that surrounds them. Artists are "set in the midst of a God-given world vibrant with a dynamic order of its own, not simply 'there' like a brute fact to be escaped or violently abused but there as a gift from a God of overflowing generosity" (Begbie 2018, p. 24). Even as artists receive an already beautiful world, this world is given to them so that through their vigorous engagement with it they can create something that "anticipate[s] by the Spirit the *shalom* previewed and promised in Christ" (Begbie 2018, p. 143). Although God created a world with order and loveliness inherent in it, God is also working in it to redeem it and bring it to its eschatological consummation, and artists engage lovingly and vigorously with the world to participate in this divine enterprise.[5]

Like Begbie, Rowan Williams argues that art-making participates in God's purposes and echoes God's way of interacting with the world. Williams constructs a theological account of art-making by meditating on Jacques Maritain's statement that "things are not only what they are", and that they "give more than they have".[6] Artists, Williams maintains,

are propelled by a desire to explore and present this excess embedded in all created things; the poetic and artistic process "blurs the conventional boundaries of perception—not to dissolve the actuality that is there but to bring out relations and dimensions that ordinary rational naming and analysing fail to represent" (Williams 2005, p. 28). In doing this, art "'dispossesses' us of our habitual perception and restores to reality a dimension that necessarily escapes our conceptuality and control". Art "makes the world strange", and thus relates in some way to the sacred, "to energies and activities that are wholly outside the scope of representation and instrumental reason" (p. 38).[7] In making art, artists perceive the materials of the world as offering more than can be grasped in one encounter, and so they create a further thing, out of these materials and in response to this encounter, that allows that deeper life to continue in another mode (p. 149). But this "making-other" is an echo of what is always already happening in things; even as the artist responds to the materials given and to the life hidden within them, that very life "is a shedder of forms, dispossessing itself of this or that shape so as to be understood and remade" (p. 153). The vocation of humans involves facilitating this process, "drawing out what is not yet seen or heard in the material environment—but not solely in exploiting it for use but in facilitating its constant movement from one material form to another, its generative capacity" (p. 154). This human activity allows the material world to move toward the eschatological fulfillment which is God's intent for it; hence, "the world's reality is always asymptotically approaching its fullness by means of the response of the imagination" (p. 154). Williams's description of art-making suggests that the material world needs the imaginative response of humans to become fully what it is; the material world needs artists who respond to what is given by transmuting it into another form, making it other and so furthering the material world's generativity.

This consistent pattern of "making other" that is embedded in the world itself, and that humans (and especially artists) facilitate, ultimately points to and echoes God's own creativity, the "making other" which is God's mode of creating the world and continuing to hold it in being. This "making other" is like God's act of creation in that it is not principally relating to things in a functional way, but rather setting them free to be independent of the creator, to have their own life. "Making-other" is an act of "dispossession, disinterested love", in that it entails a withdrawal that allows that which is made to exist in its own integrity and difference from the maker (Williams 2005, p. 161). Williams is thus arguing both that art-making reveals what true holiness is, what God's own life is like, and also that Christian theological claims about the nature of God suggest some of the norms that ought to govern art-making. In this way art-making contributes to theological understanding, and theological frameworks shape the making of art. While Williams does not use Begbie's language of redemption and restoration, Williams, like Begbie, holds that art-making contributes to the fulfillment of the world's purpose, which is ultimately God's purpose for it.

Begbie's and Williams's accounts of human artistry as an integral part of relationship with God and participation in God's purposes suggest parallels with the work of preachers. Both Begbie and Williams posit human art-making as an action that eminently reveals who God is and how God acts in the world. This is the task of preachers as well, and hence exploring how artists undertake this task can help preachers better understand what they are doing and how to do it. Like artists, preachers are "priests" or "secretaries" of creation, called to give a response of praise to the world that has been given them. Like artists, too, preachers are not creating *ex nihilo* but are working with the materials given to them, and in response to a created world already given them. Preachers eminently fulfill the vocation of all humans, to make articulate the praise that all of creation offers the Creator. At the same time, preachers, like artists, take the materials given to them and work with them, participating in God's redemptive work of drawing the world toward its eschatological fulfillment.

Preachers perform this work in the same way artists do, by immersing themselves in the reality that "things are not only what they are", and that they "give more than they

have".[8] Preachers are tasked with proclaiming this excess, this depth dimension of existence that escapes ordinary perception, conception, and control, and which is the domain of the sacred. They too are called to question everyday ways of perceiving the world, and to describe patterns and dimensions of existence that elude rational analysis. Preachers must "dispossess" themselves and their hearers of habitual ways of seeing the world so that they can perceive the divine, which eludes representation and instrumental reason. Preachers are artists in that, like artists, they are summoned into the deeps, to tune their awareness to things being more than what they are and giving more than they have, and to seek ways of rendering this excess so that others can also perceive these elusive and sacred dimensions of existence.

In rendering this excess and the wonders of these depths, preachers make the world strange to their hearers, disrupting a surface familiarity with fecund marvels, revealing how strange the world actually is. This disruption of the familiar, as Lisa Thompson notes, can often be achieved by using and then reinterpreting familiar language (Thompson 2018, p. 59). Even while they make the familiar strange, preachers, like artists, are also tasked with making the strange familiar, or at least recognizable. They do so by pointing out connections between seemingly disconnected phenomena, such as sacred texts and contemporary life. By going into the depths, they discover and can relate to others the relationships between these phenomena. The artist's work involves creating wholes out of fragments, thus making or disclosing meaning. Preachers do this, too, when they take fragments of texts and human experience and show how, in their depths, they are intimately related. As Thompson notes, moreover, the juxtaposition of the familiar and the unfamiliar has a subversive edge, because it can "expand and interrupt tradition" in ways that allow the experiences of the marginalized to be made central (Thompson 2018, p. 59).

Homileticians who explore this meeting of preaching and the arts describe how artistry in preaching opens the sermon to divine mystery. For instance, in exploring the boundaries of the sermon genre, Donyelle McCray notes that crossing this border often involves the mingling of the arts with sermons. One benefit of such experiments is to create an experience of wonder and mystery in the preaching event. Sermons consist less of "telling, explaining, and informing", and more of "stirring, imagining, reinventing, and eliciting" (McCray 2021, p. 12). Sunggu Yang, whose esthetic homiletic explores how an understanding of the preacher as artist and sermons as artistic can shape preaching in concrete ways, suggests that preachers can model their preaching after various kinds of artworks, such as film, fashion, and architecture. For instance, preachers can preach sermons in a Cubist style, rejecting the pursuit of one single meaning in texts, and developing a multi-perspectival approach to interpreting and proclaiming Scripture (Yang 2021, p. 28). Such an approach, Yang maintains, preserves the essential mystery of God and allows hearers to encounter that mystery. Likewise, an architectural sermon creates sacred spaces for esthetic meetings with the divine, rather than offering one single thesis or meaning (Yang 2021, p. 42). These comparisons of preaching to diverse art forms suggest how the artist-preacher can plumb the depth dimension of existence so as to make these deeps palpable to others.

### 3. The Role of the Self in Art-Making and Preaching

Applying accounts of Christian artistry to preaching not only discloses the similarities between sermon creation and art-making but offers a response to the concern that construing preachers as artists risks making preaching into an exercise in narcissistic self-expression without regard to external authority or community. Both Begbie and Williams seek to construct visions of the Christian artist that contest this depiction of the artist. Begbie argues that a Christian view of art counters the view that "the essence of creativity is autonomous self-expression and self-assertion . . . abandon[ing] the notion of obedience to any external agency or norm" (Begbie 1991, p. 178). The claim that artists create in response to and obedience to the God who creates and redeems the world construes art-making as responsible to God, to society, and to creation itself. Williams disputes the depiction of art-making as narcissistic by arguing that art-making is an act of dispossessing love,

echoing the way that God creates. In creating the world, God creates something other than Godself and allows it to exist in its own integrity and difference from the maker (Williams 2005, p. 161). Artists likewise engage in "making-other", withdrawing from that which they create so that it can be separate from themselves. Seeking the good of what is made, rather than imposing one's own personality on it, is one way that artists express this dispossessing love for what they create.

Preaching, too, can be seen as an expression of dispossessing love. Like artists, preachers exercise such love in their preaching, in part by attending to the good of what is made, in this case the sermon itself, rather than imposing their own personality upon it. As Williams points out, distortions in art arise when the personality of the artist distractingly obtrudes itself into the artwork; likewise, in preaching, distortions arise when the personality of the preacher is so dominant that the truth of the Gospel message is obscured (Williams 2005, p. 150). Instead, the preacher needs to be fully present in her preaching, but present in a way that sets her ego aside, so that the truth of the work itself, the Scripture to which it bears witness, and God's presence in the midst of this meeting can become apparent. The creative process facilitates this setting-aside of the ego, as it calls for a fierce attention to the world being rendered in the artwork and to that which is being made; this absorption in the work supersedes any impulse toward self-promotion, and sometimes leads to a forgetfulness of the self entirely. This setting-aside of the ego also allows space for the otherness of the hearers to be incorporated into the sermon event, because the preacher lets go of the sermon in giving it, thereby creating space for the hearer to enter into it. In doing so, the preacher allows preaching to be a truly communal event.[9]

Even as the preacher dispossesses herself of the sermon and sets it free to be other than herself, understanding the preacher as an artist highlights the way that preaching is also intrinsically an act of self-discovery. Williams argues that the love involved in creating is also intrinsically a process of self-discovery as well as a setting-free of the created object to be itself. In making, the artist uncovers what is generative in herself, as well as her own "unfinishedness". Hence, "the most profoundly free action human beings can take in relation to their identity, the action that most fully realizes the image of God, in theological terms, is to elect to discover and mould what they are in the process of 'remaking' the world", in a love that is both different from and in some ways like's God's love, inasmuch as it involves dispossession (Williams 2005, p. 165). The artist imagines a world and also imagines herself, projecting forward an identity that is in motion toward an unseen goal (p. 167). The preaching life is likewise an ongoing process of self-discovery. Even though sermons ought not to be "about" the preacher, they are nevertheless a key vehicle for the preacher's ever-deepening self-formation. Indeed, one of the ways that a preacher remains alive in her preaching is that she is always coming to know God and herself better by engaging in the task of proclamation. As John Calvin argues, self-knowledge and knowledge of God are inescapably related, and in the creative act of preaching, the preacher continually explores this relationship (Calvin 1960, pp. 35–39). This also means, however, that the preacher who does not want to know herself, who avoids self-knowledge, will be limited in how deeply she can know God and, ultimately, in what she is able to say.

## 4. Imagination and the Real in Art-Making and Preaching

Artists explore the depth dimensions of the world through the use of the creative imagination. Some Christian suspicion of art-making is rooted in a sense of the moral ambiguity of the human imagination, understood as a faculty prone to generating evil thoughts that can lead to evil actions. As Trevor Hart notes, the capacity to imagine something other than current reality is at the root of Adam and Eve's rebellion against God in Eden (Hart 2014, pp. 48–49). After this original fall from grace, the Bible describes the "imaginings of [human] hearts" as tending toward evil (cf. Gen 6:5). Despite the capacity of the imagination to break faith with God's vision and to incline toward fantasy and deceit, Hart insists that the imagination is a faculty of genuine knowledge and is particularly crucial in theological knowing: "theology is a human activity in which the quotient of

imagination is set extraordinarily and necessarily high" (Hart 2014, p. 11). The use of the imagination is necessary in theology because God is utterly different from us, and thus the only way we can speak of God is through image and metaphor. Writing about the work of David Brown, Douglas Hedley argues that Brown's work seeks to demonstrate that "deeper forms of truth can only be appropriated through the imagination" (Hedley 2012, p. 80).

If the imagination is necessary for artists, it is yet more crucial for preachers, whose task is to speak of God, and thus must rely on image and metaphor to describe the One who is utterly different from us. Ellen Davis, defining and defending the use of the imagination in preaching, describes the imagination as "the interpretive faculty by which we relate to that which is strange, not fully known, or not immediately present to us" (Davis 1995, p. 253). The imagination also has "a mystical dimension, in the sense of involving us, to a greater or lesser extent, in an apprehension of the totality of life, which is essentially mysterious" (p. 254). These qualities of the imagination make it especially suitable, and, indeed, necessary, in speaking of God, whom Scripture describes less in concepts and more in images—the rock, the mother hen, the dove, the fire. Preachers likewise inevitably draw on their imaginations to describe God, to enter into the conceptual landscapes of biblical texts, and to call hearers toward God's purposes. As Cleophus LaRue describes it, "before, during, and after formal preparation for preaching, the imagination is there pushing, cajoling, pressing its case, speaking into the preacher's ear and trying to gain a hearing if only the technical will occasionally yield to the poetic" (LaRue 2011, p. 73).

Because the imagination draws us to imagine that which we have not yet fully attained and do not fully understand, it is a powerful vehicle for transformation, both personal and political. It is for this reason that Walter Brueggemann describes the "prophetic imagination" as the capacity both to unmask the death-dealing realities of the world as it is, and to offer a vision for the world made whole that is God's intent (Brueggemann 2001, pp. 39–41, 59–60). Preachers are involved in an imaginative project: to cast a vision of the world as God sees it and as God desires it to be, so as to draw people from the world as it is now to the world as God intends it. Otis Moss III notes the power of the imagination to provoke change in his description of Blue Note preaching as an artistic enterprise. Blue Note preaching, which Moss defines as the capacity to preach about tragedy but not fall into despair, arose powerfully in the context of African American enslavement. In such contexts of oppression Blue Note preaching relies on the imagination, because it is "the creation of an alternative world and an alternative consciousness", which allows hearers to envision liberation, "to be carried away to a land conceived of only in their imaginations" (Moss 2015, p. 26).

Although the imagination allows preachers to draw hearers to envision a world of justice which we cannot yet fully perceive, imagination is nevertheless rooted in the real. Indeed, the key difference between imagination and fantasy, Hedley argues, is that the imagination illuminates reality, prompting transformation, whereas fantasy indulges in make-believe and does not challenge us to change (Hedley 2012, p. 86). Because the imagination is rooted in reality, to explore the depth dimensions of existence is also, paradoxically, to attend with scrupulous attention to the things of this world, that which gives itself to one's perception. Even though the artist seeks to express the "invisible structures" of the world, she does so by attending to "the actuality of objects" (Williams 2005, p. 18). This in part echoes the fact that only God creates *ex nihilo*, and only God grasps things as pure ideas; humans, on the other hand, create out of what is given, and it is only in relationship to the created world and objects in it that they are able to perceive the depth dimension of existence. Flannery O'Connor urges Christian writers to "do justice" to the physical world, which means grounding their writing in concrete, observable reality (O'Connor 1969b, p. 148). The writer, she argues, "cannot move or mold reality in the interests of abstract truth", and instead must recognize that "what-is is all he has to do with; the concrete is his medium; and he will realize eventually that fiction can transcend its limitations only by staying within them" (O'Connor 1969b, p. 146).

This advice to writers is apposite for preachers as well. Preachers, too, must grapple with concrete reality instead of talking only about ideals. It is through concrete and

particular lived experience that humans, both preacher and hearers, actually come to know the divine and to see the excess in the nature of things that Williams describes. Preachers, like artists, do not come to see this excess in the abstract, but only through that which is most concrete. Thompson urges preachers to "make the faith-story concrete", since "renaming and reassigning the faith-story through everyday words and associations helps locate and construct theological meaning in ways that remain in close proximity to lived experience" (Thompson 2018, p. 164). This attention to lived experience is what makes Christian faith claims credible, especially to the lives of the marginalized. It is for this reason that Mitzi J. Smith describes womanist biblical interpretation as making lived experience, notably that of black women and their communities, "a point of departure, focal point, and an overarching interpretive lens for critical analysis of the Bible and other sacred texts" (Smith 2015, p. 4).

In contrast, the avoidance of concrete reality signals an unwillingness to grapple with the mess of this world, perhaps rooted in a sense that being people of faith means that there are certain things preachers cannot accept or should not see. However, as O'Connor argues, being a person of faith ought to enlarge rather than narrow one's perception; it ought to give one freedom to observe the world without restraint, not to avert one's eyes from any reality, in the trust that all of reality, even the most depraved, is within the scope of God's saving action. As Williams puts it, the Christian writer's task is "to forget or ignore nothing of the visually, morally, humanly sordid world, making nothing easy for the reader, while doing so in the name of a radical conviction that sees the world being interrupted and transfigured by revelation" (Williams 2005, p. 100). Preachers sometimes make the mistake of glossing over the suffering and evil in the world in order to proclaim God's grace, but this is not credible; the vocation and practices of artists are a reminder to avoid this mistake.

Relatedly, O'Connor cautions Christian writers not to allow the dogma and tradition of the church to short-circuit their own perceptions and creativity:

> The sorry religious novel comes about when the writer supposes that because of his belief, he is somehow dispensed from the obligation to penetrate concrete reality. He will think that the eyes of the Church or of the Bible or of his particular theology have already done the seeing for him, and that his business is to rearrange this essential vision into satisfying patterns, getting himself as little dirty in the process as possible...But the real novelist, the one with an instinct for what he is about, knows that he cannot approach the infinite directly, that he must penetrate the natural human world as it is. (O'Connor 1969a, p. 163)

It is possible for artists to be blinkered by religious tradition and not set free by it, and this condition causes artists not to really see the world, not to fully engage with concrete reality. Christian dogma and tradition can limit the artist to seeing only what seems to fit within these frameworks, but as O'Connor cautions, this tends to lead to shallow and predictable work. Indeed, it is because artists allow themselves to be constrained by the structures of Christian tradition that so much second-rate Christian art is created, since abstract ideas and strictures have been allowed to dictate the boundaries of the creative process, rather than an encounter with concrete reality providing the basis for creative work.

As a way of counteracting this tendency, Williams, echoing O'Connor, argues that the central Christian mystery, God's self-revelation in the life, death, and resurrection of Jesus, means that all of created reality has been deemed worth God's dying for, and thus nothing can be left out of the purview of the Christian artist. This truth suggests not only that the artist leaves nothing out of her awareness when creating art, but also that the artist must focus solely on the integrity of what is made, and not interrupt its internal logic by introducing statements of meaning that are external to the work itself: "If the writer urgently wants to lay bare a moral universe or a dogmatic structure, she has to do so exclusively in the terms of the work itself, not by introducing a moral excursus or by holding back because of possible undesirable results in a vulnerable reader" (Williams 2005, p. 96). The meaning of the work must be carried inside the work, not appended to it from outside. Williams notes that propaganda is work that has a message or purpose separable

from the work itself, such that the work becomes a means to an end. Art, Williams argues, is the one "intransitive" action of human beings, an action that has no ulterior purpose, but is an end in itself (p. 53). Art to which dogmatic or moral meanings are externally attached rather than being intrinsic to the work itself is art that has ceased to be an end in itself but serves some other purpose, in which case it is compromised as art.

It could be argued that preachers are not as free as artists to set aside the constraints of dogma and tradition and focus purely on what they see, that preachers need to proclaim the Good News, and must bring all of their perceptions and insights into alignment with this requirement. Indeed, one of the chief objections to viewing preachers as artists is that artists are not bound to external authority, whereas preachers are. However, Williams's guidance to attend only to the good of what is made urges preachers to read both texts and contexts fearlessly and openly, and to be truthful about what they perceive. As Anna Carter Florence observes, preachers often feel constrained to wrestle a meaning out of texts that conforms to received dogma, but she points out that this insistence can do violence to the text. She describes how her students can be entirely free and creative when exegeting texts, but as soon as they focus on preaching those texts, their openness disappears as they search for the "meaning" of the texts: "The students start hacking away at the text until they can grasp a piece of it to hold up, while announcing, 'Let me tell you what this passage [insert any biblical pericope] *means:* it *means* that everything will be fine if we only have faith!'" (Florence 2004, p. 94; emphasis original). This response to the text short-circuits what the text itself may be saying in favor of repeating a pre-determined dogma. Instead of doing this, Florence urges her students to *explore* the text rather than *explaining* the text, simply to preach the text rather than trying to make it logical, relevant, or doctrinally correct. Rather than being the ones with the answers, preachers are "*the ones who will look, and the ones who can describe exactly what we see in our life in the text*" (Florence 2004, p. 102; emphasis original).

This description of the preacher's task echoes O'Connor's advice to writers to perceive concrete reality as honestly as possible and not to curtail their vision based on received doctrine. As Williams suggests, preachers can focus purely on the integrity of the sermon itself, rather than appending statements of meaning that are extrinsic to the work itself. If the good of what is made in a sermon is a truthful reflection on what is seen in the text and in life, that ought to be the preacher's sole focus, rather than extracting a meaning from this seeing. The meaning of the sermon must be carried inside the sermon itself, inside its truthful vision of the text and the world, rather than added onto it from outside. The artist must reckon with what concrete reality actually presents her, and only in this reckoning disclose the depth dimensions hidden in these realities. As womanist biblical scholars, among others, point out, the litmus test for this truthful seeing is that it prioritizes the life experience of the dispossessed (Smith 2015, p. 4; Thompson 2018, p. 85).

## 5. Unknowing in Artistry and Preaching

One outcome of this courageous and honest perception of both Scripture and the contemporary world is that it may compel preachers to abandon a stance of certainty and to take up a posture of unknowing. As Florence notes, to see both world and texts truthfully, and to describe honestly what one sees, may lead the preacher to say, "I don't know" (p. 104). The preacher's task is to preach "*only what we have seen and heard in the text—which is sometimes the emptiness of what we have not seen; at least, not yet*" (p. 105; emphasis original). This unknowing is intrinsic to the artistic process; the artist, by definition, does not know where he is going when he begins to create. This is one of the key differences between art and craft: craft involves following a template, whereas art does not. Art-making is a venture into the unknown, and for preachers to embrace their practice as an artistic one is to enter into this realm of the unknown and the risky stance of unknowing. This posture keeps preachers from short-circuiting both text and life, allowing them to create a proclamation that bears truthful witness to the world that God has considered worthy to die for. This

attitude, that of the artist, moves preachers beyond superficial answers and clichés into a deeper wisdom.

As ventures into the unknown, both preaching and art-making are finally efforts to capture that which can never be fully captured. As Williams points out, art limps from the encounter with what cannot be named (Williams 2005, p. 21). Likewise, the beauty we encounter in the world makes us want to create something beautiful, and yet the beauty we receive and perceive always outstrips what we can create. T. S. Eliot, describing the struggle to write poetry, writes of the "intolerable wrestle with words and meaning", in which "every attempt/Is a wholly new start, and a different kind of failure", the inevitable failure to say what he most wants to say (Eliot 1943, pp. 26, 30). Yet, as Begbie notes, these inevitable failures lead the artist to an ever-deeper praise of the beauty she did not and cannot create but can only receive. In the very effort of trying and failing to grasp the beauty of the world in our art-making, we ourselves are grasped by that which exceeds us. As I try and fail to capture, through drawing and painting, the beauty of the magnificent black locust tree outside my window, I surrender myself to the tree itself, to a majesty that transcends my rendering of it. In Martin Buber's terms, I have an I-Thou encounter with the tree, but this encounter takes place because of my intense wrestling to capture its essence on paper, a wrestling in which I lose but am blessed (Buber 1958, pp. 7–8).

This wrestling to describe that which cannot be fully named is intrinsic to preaching as well. As speech that aims to speak truthfully of God, preaching is, in one sense, always an impossible task, always doomed to failure. God is absolutely other, and cannot be fully grasped or described in human language. Hence, preaching, like art-making, limps from its encounter with that which cannot be named, as Jacob limped after his wrestling match with the mysterious stranger who both wounded and blessed him (cf. Gen 32:22–32). Rather than denying this incapacity, or simply giving up, preachers are called to continue to wrestle with what cannot be said, keeping their words as close as possible to the unsayable. By doing this, preachers also adopt a posture of humility that counteracts the risk of egotism in preaching.

## 6. The Courage of the Artist and Preacher

These various aspects of art-making—the dispossession of the artwork, the ongoing self-discovery and self-transformation of the artist, the fearless encounter with the realities of this world, the imaginative projection forward into a new world, the unknowing that attends the artist's process, the inevitable failure to realize one's vision in one's art—all point to the necessity of courage as the essential attribute of the artist. Art-making, as a venture into the unknown, into the depths of the world and the depths of the self, is inevitably an encounter with fear. This fear can be fear of the depths themselves, of exploring aspects of life that may be frightening or overwhelming. This fear can also be a fear of one's own incapacity as an artist. A volume aptly entitled *Art and Fear* explores this dynamic, pointing out that self-doubt often attends the process of art-making (Bayles and Orland 1993, p. 13). Artists typically fear that they will not be able to create, or that they have nothing to say, or that their work will be inadequate. In part, such fears arise because artists are, by definition, treading an unknown and not a pre-determined path. These fears also arise because there is an inevitable failure built into art-making, as an enterprise in attempting to capture that which cannot be captured.

Comparing preachers to artists suggests that courage and risk-taking are essential to the art of preaching as they are to other forms of art-making. If preachers embrace the artist's call to follow a path without knowing where it will lead, there is risk involved in this open-ended exploration. Relatedly, there is risk in saying "I don't know", concerning the meaning of either life or texts. There is risk involved in allowing this open-endedness to challenge the boundaries of theological or hermeneutical orthodoxy, in accepting O'Connor's challenge to the Christian artist not to let the doctrine of the Church constrain what the preacher will see or say. As Donyelle McCray points out, preaching is often a highly censored activity, in which what preachers can say feels quite constrained (McCray 2019, p. 1). The artist

metaphor, by allowing preachers to resist censoring themselves, to think outside of the box and color outside of the lines, offers a sometimes frightening freedom. There is risk involved in the ongoing self-discovery that the artist-preacher undertakes. There is risk in seeking to speak of God when God can never be fully captured in language. There is risk in dispossessing oneself of one's creation, setting aside the ego for the sake of the goodness of what one is creating. Most profoundly, as Lisa Thompson argues, "every sermon is an act of risk-taking if we—every preacher, *outsider* or not—are faithful to our preaching ministries. For we are attempting to say something fully life sustaining in a world that often affirms death over life" (Thompson 2018, p. 176, emphasis original).

The preacher-as-artist metaphor makes clear that it is the willingness and courage to take these risks that gives life to one's preaching. A good litmus test for the aliveness of one's preaching is the extent of risk one is taking in it. Bayles and Orland maintain that when artists look at their artwork, their art will tell them where they are holding back and where they are fully engaging (Bayles and Orland 1993, p. 49). The metaphor of preacher as artist likewise challenges preachers to assess their preaching to see where they are holding back, and to consider the risks in preaching that God might be calling them to take.

### 7. Conclusions

Considering preaching as a form of artistry, and understanding artistry as intrinsically connected to God and God's purposes, sheds light on why the art-making exercise in my introductory preaching class generates so much delight and insight. The exercise allows students to express the innate urge to create that is in all of us as made in the image of the Creator. Making art allows them to explore unconventional ways of perceiving the world, drawing forth unexpected connections that allow for the discovery and articulation of deep and unexpected truths about themselves, the Scripture texts, the world, and God. They exercise the freedom to respond to and interpret Scripture without constraint. They welcome the invitation to leave things open-ended, to say what they do not know instead of having to pin things down. In accepting this permission they encounter fear—lest they should fail, lest they have nothing to say, lest they should stray too far outside the bounds of orthodoxy—but at the same time they revel in the invitation to take these risks, discovering their courage in the process of doing so.

Understanding the benefits of an art-making assignment as part of the sermon creation process suggests not only the value of this assignment specifically. Analyzing why this assignment is beneficial suggests the broader value of understanding and practicing preaching as artistry. As preachers claim their vocation as an artistic one, they gain access to the creativity, courage, insight, and freedom that art-making invites and requires. They exercise the intrinsic human need to create, which has the capacity to participate in God's redemptive purposes and be an echo of God's creative love. In the sermons they create, they then invite their hearers to participate in similar acts of loving and redemptive creativity.

**Funding:** This research received no external funding.

**Institutional Review Board Statement:** Not applicable.

**Informed Consent Statement:** Not applicable.

**Data Availability Statement:** There are no additional data available for this research.

**Conflicts of Interest:** The author declares no conflicts of interest.

### Notes

[1]  Sunggu Yang advocates similar artistic practices in the homiletics classroom, arguing that creating art in response to Scripture texts leads to a "holistic-aesthetic experience of the text" that draws students closer to the *mysterium tremendum* to which the text bears witness (Yang 2021, p. 13).

[2]  For a further discussion of the relationship between art and idolatry, see Vincent (2016). Vincent argues that idolatry is more properly located in the intent of the maker of art than in the recipient's attitude toward the artwork (p. 393).

3  See also Deborah Sokolove's helpful historical summary of the ambivalent and complex relationship between Christianity and the arts, especially the place of the arts in Christian liturgy (Sokolove 2022).

4  Thomas Long suggests this critique of the preacher as artist in his review of the disadvantages of the storyteller/poet metaphor for the preacher. This metaphor, Long argues, can place too much attention on the linguistic beauty of the sermon or the captivating quality of the storyteller, rather than on the gospel (Long 2016, pp. 48–49). Richard Viladesau, likewise, in arguing that preaching should be artful, acknowledges that this understanding risks minimizing "the intrinsic power of the message or the efficacy of God's Spirit", but maintains that this potential risk is countered by affirming that "God's work and human agency are directly, and not inversely, related" (Viladesau 2000, p. 178). The worry about preachers being artists goes back at least to Karl Barth, who cautions that preachers are to present the Scripture as purely as possible, intruding as little of their own personalities as possible, and that they "should not practice our own arts with the Word, because the Word that ought to be spoken will not be heard" (Barth 1991, p. 94).

5  Begbie also points out that beauty itself must be redefined by Christian artists, in light of the cross of Christ. If the cross is God's fullest self-revelation, and hence the fullest revelation of divine beauty, then beauty must encompass suffering, evil, even what would be considered ugly by worldly standards (Begbie 1991, p. 158). Ivan Dodlek similarly stresses that Christian beauty, and hence Christian art, is "kenotic", echoing God's kenosis, and thus including the "imperfect, fragmentary, and painful in its artistic expressions" (Dodlek 2023, p. 8). See also David Bentley Hart's extensive treatment of Christian beauty (Hart 2004, pp. 153–248).

6  Williams (2005), p. 26; quoting Maritain (1953), p. 127.

7  For a similar argument about the capacity of art to reveal the mysterious depths of existence, see Dodlek (2023), p. 4.

8  See note 6 above.

9  Creating an integral role for the listener in preaching, and hence making preaching a communal event of meaning-making, has been a feature of recent homiletical theology. See, for instance, McClure (1995) and Rose (1997). Meaghan Burke suggests how art-making can likewise be a communal activity, and in this sociality be a form of prayer, binding communities together and in so doing revealing their unity in Christ (Burke 2023).

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
