# Peer review of "The Preacher as Artist: An Exploration of Sermon Creation as Art-Making"

_religions, doi:10.3390/rel15050604_

Round 1

Reviewer 1 Report

Comments and Suggestions for Authors

The article is making a contribution within a field that has been explored over many years. The article is at times difficult to read due to the argument jumping from one important issue to another.  Coherence between paragraphs could enhance easier reading. The author/s are encouraged to do smaller revisions and to publish the research. However, it is my opinion that thoughts on the dynamic between hearing as seeing and further elaboration on imagination and memories could further enrich the current research.

Author Response

Thank you for the review of this essay. I have responded to your review by adding sentences at the beginning and endings of several sections of the paper to make the transitions clearer. I hope to incorporate your thoughtful suggestions on the relationship between hearing and seeing, and between imagination and memory, in my further work on this topic.

Reviewer 2 Report

Comments and Suggestions for Authors

This essay is nuanced and insightful in its theological argumentation, and it is helpfully practical in its concrete pedagogical advice for homileticians. I commend the author for their good work, and I firmly believe this essay will make an important contribution to our field.

At the same time, I think the essay could be strengthened by more thoroughly engaging non-white authors. Again, I think this essay is brilliantly written and makes an important contribution to our field. My only concern is that its influence might be limited because its primary conversation partners are white.

Author Response

Thank you very much for this review of my essay. Your prompting to engage non-white authors is much appreciated, and I have done this in my revised essay. In particular, I have incorporated the work of Cleophus LaRue, Otis Moss III, Donyelle McCray, Lisa Thompson, and Mitzi Smith. I believe the conversation with these authors has strengthened this essay, and I thank you once again for this important feedback.